# SSONN: Self-Scaled Optimized Neural Network

## Abstract

Current approaches to lightweight neural network design face a fundamental trade-off: reducing model size inevitably compromises accuracy. Distillation and pruning are the most commonly used methods, they require an initially over-parameterized pretrained architecture that increases computational costs while training. This work introduces a novel Self-Scaled Optimized Neural Network (SSONN) method that eliminates the need for redundant initial models. Instead of following a *train-then-compress* paradigm, SSONN starts with a single linear layer and dynamically increases its complexity during training through adaptive reverse pruning. Rather than removing redundant parameters, the algorithm selectively adds nodes and connections only to critical places to improve task-specific accuracy. Similar methods that perform dynamic expansion of neural network architecture rely on well-known architectures or their parts, whereas the proposed method is independent of existing architectures. Experiments on classical datasets (MNIST, Fashion-MNIST, and Unseen NAS datasets) demonstrate that SSONN achieves accuracy comparable to state-of-the-art models while using 10 times fewer parameters. Furthermore, the method outperforms traditional training approaches in computational efficiency and enables flexible deployment in resource-constrained environments. These results highlight the potential of the suggested expansion strategy over the reduction approach for creating efficient and adaptive deep learning models. The code is available at [1].

## 1 Introduction

The growing demand for deploying neural networks in resource-constrained environments has intensified the need for lightweight yet accurate models. Traditional model compression techniques – including knowledge distillation (Hinton et al., 2015b) and pruning (Han et al., 2015) – have become the core tools for reducing computational complexity. However, they rely on a counterintuitive workflow: starting with an over-parameterized model, even for a simple task, and consequently eliminating its redundancy. This approach not only demands significant surplus computational resources, but also leads to an unavoidable accuracy-efficiency trade-off, where aggressive compression tends to irreversibly degrade model performance (Blalock et al., 2020). Compression processes are often deterministic and task-agnostic, therefore, there is a risk to remove features critical for specialized applications.

Neural Architecture Search (NAS) automates the design of efficient architectures but is hindered by high computational costs and rigid search spaces (Elsken et al., 2019). Moreover, once an architecture is chosen, it becomes fixed and cannot be changed for a specific task. Dynamic architecture adaptation methods, including self-expanding networks, offer a promising alternative by incrementally growing compact models. However, existing approaches, such as Firefly (Wu et al., 2021), GradMax (Evci et al., 2022), and SENN (Mitchell et al., 2024), often depend on fixed schedules or require extensive hyperparameter tuning, which limits their flexibility.

In this work, we propose a paradigm shift. Instead of compressing or using existing models or their components to meet task-specific accuracy requirements, we apply the method that allows a compact network to expand. Our Self-Scaled Optimized Neural Network (SSONN) begins with a

---

[1]The link is removed due to blind review, the code can be found in supplementary material.

simple architecture and selectively adds neurons and connections only where necessary during training. Based on adaptive pruning techniques, SSONN employs a gradient-driven criterion to identify potentially suitable network regions, iteratively expanding capacity in a targeted manner. This approach eliminates the need to pretrain over-parameterized models, reducing initial computational costs according to task complexity.

We have evaluated SSONN on standard NAS benchmarks (MNIST, Fashion-MNIST, and Unseen NAS datasets (Geada et al., 2024)), demonstrating three key advantages. SSONN demonstrates its efficiency matching the accuracy of dense baselines while using fewer parameters on average. The model dynamically adapts its architecture, which minimizes the use of computational resources at each training stage. The network architecture does not add new neurons into existing layers or insert new fully-connected layers between the existing ones; it expands by adding new sublayers and trains them. This provides the model's automatic scaling, which is beneficial for more precise architectural modifications.

The present work contributions are the following:

- We propose a novel expansion-centric methodology that challenges the traditional *train-then-compress* paradigm;
- We show that gradient-guided dynamic growth outperforms classical pruning while reducing initial resource requirements;
- We implement an approach that achieves a more optimal lightweight architecture independent of existing architectures or search spaces.

This work bridges the gap between model efficiency and adaptability, suggesting a pathway to sustainable deep learning in resource-limited settings. By rethinking network design as an adaptive process rather than static optimization, SSONN promises creating compact, task-specific architectures without sacrificing accuracy.

## 2 RELATED WORKS

Current approaches to neural network optimization can be divided into three key areas: model compression methods, Neural Architecture Search (NAS), and dynamically adaptive architectures. Each of them has its unique advantages and limitations, important to consider when developing new methods.

### 2.1 TRADITIONAL COMPRESSION TECHNIQUES

Historically, **Pruning** (Vadera & Ameen, 2021) is a significant approach to enhance model efficiency by removing redundant neural network parameters. For example, estimating the contribution of neurons to the final loss using the Taylor expansion was proposed (Molchanov et al., 2019). This approach enables scalable pruning across all network layers without requiring layer-specific sensitivity analysis; it achieves up to $30\%$ parameter reduction with only a $0.02\%$ accuracy loss. The Lottery Ticket Hypothesis (Frankle & Carbin, 2019), another significant contribution, demonstrates that a fully connected network can be trained, pruned to remove up to $75\%$ of its parameters, and then reinitialized with the original weights for retraining. Remarkably, the resulting sparse network achieves the accuracy comparable to the original dense model but with significantly fewer parameters. Furthermore, the Dynamic Network Surgery method (Guo et al., 2016) introduces a dynamic pruning strategy that allows pruned connections to be reinstated if they become important during subsequent training. By continuously evaluating and adjusting the network structure, this method achieves impressive compression rates without accuracy loss: $108\times$ for LeNet-5 and $17.7\times$ for AlexNet on benchmark datasets. Despite all the advantages of pruning, to obtain a more lightweight model, the initial model with a large number of parameters needs costly substantial training.

Popularized by (Hinton et al., 2015a), **Knowledge Distillation** method transfers knowledge from a large teacher model to a compact student model, typically by minimizing the Kullback-Leibler Divergence (KL-Div) between their output distributions. Despite its success, it relies on the training of a large teacher network, which can be computationally expensive. Recent advancements (Tian et al., 2019) have sought to address KL-Div's limitations, such as its inability to capture cross-category

relationships and its poor handling of non-overlapping distributions in intermediate layers. For instance, (Lv et al., 2024) proposes using the Wasserstein Distance (WD) as an alternative to KL-Div for knowledge distillation. The innovations improve the transfer of rich inter-class relationships and geometric properties of feature spaces, outperforming KL-Div-based methods in tasks like image classification and object detection. However, even with these improvements, knowledge distillation still requires an initial large teacher model.

## 2.2 NEURAL ARCHITECTURE SEARCH (NAS)

Neural Architecture Search (NAS) has emerged as a pivotal method for automating neural network design, aiming to balance accuracy and efficiency for specific tasks. While early approaches like DARTS (Liu et al., 2018) pioneered differentiable search strategies, recent advancements have expanded NAS capabilities across search efficiency and flexibility.

**Search Efficiency and Flexibility**. Current NAS methods increasingly decouple architecture discovery from full training cycles. For instance, (Shen et al., 2024) introduces a training-free subnet search for large language models, leveraging weight importance initialization and evolutionary algorithms to outperform structured pruning baselines like SliceGPT. Similarly, (Ericsson et al., 2024) constructs a unified search space using probabilistic context-free grammar; which enables expressive architectures spanning ResNet and transformer paradigms. These works demonstrate NAS potential to reduce manual bias, yet they inherit a critical limitation – static architectures post-search. For example, Supernet Shifting (Zhang et al., 2024) optimizes global-local supernet consistency and achieves a $10\times$ speedup in cross-dataset transfers (e.g., ImageNet to CIFAR-100), but the final architecture remains fixed, unable to adapt to new data distributions without retraining.

**Limitations.** Despite notable advancements, NAS frameworks remain constrained by two fundamental challenges. First, their architectures, even for top methods like (Shen et al., 2024) and (Ericsson et al., 2024), are static and task-specific. The finite search space may miss the optimal architecture for the given data, relying mostly on SOTA designs. Second, the computational costs associated with architecture discovery remain prohibitively high. Techniques relying on evolutionary algorithms or supernet training, such as those in (Zhang et al., 2024) and (Zhao et al., 2024), demand thousands of GPU hours for search and evaluation, creating a paradoxical contradiction: methods designed to optimize efficiency become resource bottlenecks.

## 2.3 APPROACHES TO DYNAMIC ARCHITECTURE ADAPTATION

The evolution of neural network optimization methods has naturally led to the idea of self-expanding architectures. While traditional methods were mainly concerned with the reduction of network components, current approaches strive for dynamic architecture adaptation depending on input data.

Parameter addition can focus on either expanding the width or deepening the network. Cascade-Correlation (Fahlman & Lebiere, 1989) starts with a minimal network and incrementally adds new hidden units, each trained to maximize the correlation between its output and the network residual error. Fukumizu and Amari (Fukumizu & Amari, 2000) demonstrate that adding neurons can transform local minima into saddle points, helping to escape suboptimal solutions.

Recent research has focused on addressing the key questions of when, where, and what to add in terms of new parameters.

**Firefly Neural Architecture Descent framework** (Wu et al., 2021) is used for progressively growing neural networks by iteratively optimizing both parameters and architectures in a steepest descent manner. The method defines a functional neighborhood of the current network, using the Taylor approximation and greedy selection to efficiently grow the network wider (by splitting existing neurons or adding new ones) and deeper (by inserting residual layers). Applied to neural architecture search and continual learning, it achieves competitive performance on benchmarks like CIFAR-100, though it relies on computationally intensive optimization and predefined backbones, such as reduced VGG-16 or MobileNet V1.

**GradMax** (Evci et al., 2022) is a technique that uses gradient information to incrementally grow neural networks from predefined networks. This approach ensures that the integration of additional neurons preserves the integrity of previously learned representations, thereby maintaining the net-

work existing knowledge base. The initialization of these neurons is refined by applying singular value decomposition (SVD), which efficiently identifies optimal weight configurations. However, its fixed-interval expansion may not suit all tasks optimally.

**Self-Expanding Neural Network (SENN)** (Mitchell et al., 2024) employs a natural expansion metric based on the natural gradient (Amari, 1998) and a curvature approximation to determine the timing, location, and initialization of adding neurons or layers. This method starts with a compact architecture, such as a multi-layer network with minimal active neurons, and dynamically expands it based on task demands. SENN supports expansion by such techniques as skip connections for depth addition and masked neurons for efficient width growth. It also incorporates pruning to remove redundant parameters. However, its reliance on tuning numerous hyperparameters, such as expansion thresholds and curvature approximations, can complicate its application and conflict with the goal of fully automated architecture optimization.

## 3 METHOD

SSONN, a self-expanding neural network, overcomes the limitations of fixed architectures by allowing the network to adaptively increase its representational capacity in response to learning challenges. The network architecture is modified while training by strategically adding new neurons and connections. The expansion process occurs iteratively, targeting specific network regions identified through gradient-based importance metrics.

The key difference of our approach lies in creating sublayers instead of adding new neurons into the existing layers or inserting new fully-connected layers between the existing ones. This targeted expansion strategy allows for more precise architectural modifications.

From now on, for convenience, we refer to the connections between neurons *edges* and neurons themselves as *vertices*. SSONN cycles repeatedly through *edge selection*, *expansion*, *weight initialization*, *training*, and *pruning*. First, it computes an edge-wise importance metric (Sec. 3.1) and selects every edge whose metric exceeds a fixed threshold $\tau_s$. Next, for each selected edge, it inserts an intermediate neuron, connecting that neuron to the original parent and child vertices (Sec. 3.2), and initializes the new weights as prescribed in Sec. 3.2 so that the network output remains essentially unchanged immediately after expansion. Then, standard training (Sec. 3.3) resumes, allowing the newly introduced parameters to adapt via gradient-based optimization. Finally, several epochs after expansion, it prunes all connections whose importance metric has fallen below a second threshold $\tau_p$. Hyperparameters determine how often this expansion–pruning cycle is triggered during training (see subsection A.2). During training, this procedure may be executed repeatedly, with hyperparameters regulating the frequency of expansion phase initiation.

### 3.1 EDGE SELECTION

To determine which edges to replace, an evaluation criterion is required. Thus, we have designed edge selection metrics that estimate how much each edge contributes to the final prediction. Based on the values of these metrics, we select the edges to be replaced and then apply the suggested transformation.

We have experimented with several approaches, including selecting edges based on absolute weight values, the squares of weights. However, gradient-based metrics proved to be the most effective (see subsection A.3), aligning with the sensitivity analysis for neural network pruning (Enzo Tartaglione & Grangetto, 2021).

In particular, we employ two metrics, the one that computes the mean gradient magnitude and another that computes the gradient variance:

$$g_{\text{abs}}(e) = \frac{1}{M} \sum_{j=1}^{M} \left| \frac{\partial L^{(j)}}{\partial w_e} \right|, \tag{1}$$

$$g_{\text{var}}(e) = \text{Var}_{j=1,\ldots,M}\left[\left|\frac{\partial L^{(j)}}{\partial w_e}\right|\right] = \frac{1}{M} \sum_{j=1}^{M} \left| \frac{\partial L^{(j)}}{\partial w_e} \right|^2 - \left( \frac{1}{M} \sum_{j=1}^{M} \left| \frac{\partial L^{(j)}}{\partial w_e} \right| \right)^2, \tag{2}$$

where $g_{\text{abs}}$ is the mean absolute gradient value, $g_{\text{var}}$ is the mean absolute gradient variance magnitude, $M$ is the total number of mini-batches, $L^{(j)}$ is the loss evaluated on the $j$-th sample, and $\partial L^{(j)}/\partial w_e$ is the gradient of $L^{(j)}$ with respect to weight $w_e$, Var is the variance. As our experiments show (see subsection A.1), the performance of these two metrics is nearly identical. We have used the $g_{\text{abs}}$ edge selection metric as it can be computed with lower costs.

We select the edge $e$ for expansion when $g_{\text{abs}}(e) > \tau_s$. Our approach contrasts with other gradient-based pruning methods (Enzo Tartaglione & Grangetto, 2021), where low gradient magnitudes typically indicate redundant connections. We do interpret high gradient values as markers of important edges that might drive learning progress. The gradient-based criteria effectively capture the sensitivity of the loss function to changes in trainable network parameters, which confirms the motivation for our expansion strategy.

Edges with a low metric value $g_{\text{abs}}(e)$ exhibit low sensitivity: small perturbations of their weights barely change the objective, signaling that the corresponding inputs contribute little new information. In contrast, edges with a large metric value are high-sensitivity connections, as small updates there induce noticeable variations in the loss (see subsection A.3). SSONN therefore expands the network by injecting extra capacity precisely where the model is most responsive; this targeted expansion translates into measurable performance gains (see subsection A.1).

## 3.2 LAYER EXPANSION

Let $G = (V_p, V_{p+1}, E, W)$ denote the directed graph of a linear layer, where $V_p$ and $V_{p+1}$ are the sets of vertices in the $p$-th and $(p+1)$-th layers, respectively, $E$ is the set of edges and $W$ is the map of weights. Given a scalar threshold, $\tau_s$, we collect the set of *selected edges* as follows:

$$E_{\text{sel}} = \big\{ e = (v_i^p, v_i^{p+1}) \in E \,\big|\, g_{\text{abs}}(e) > \tau_s \big\}, \quad |E_{\text{sel}}| = m. \tag{3}$$

Then, for each edge $e_i \in E_{\text{sel}}$ we add an *intermediate vertex* $v_i^n$, producing three node sets:

$$P = \{v_i^p\}_{i=1}^k, \ N = \{v_i^n\}_{i=1}^m, \ C = \{v_i^{p+1}\}_{i=1}^s,$$

where $P$ is the set of selected parent vertices, $N$ the newly added intermediate vertices, and $C$ the set of selected child vertices, $0 \le k \le |V_p|$, $0 \le s \le |V_{p+1}|$.

Now we want to create a sublayer consisting of vertices from $N$. By doing so, we delete all edges from $E_{\text{sel}}$. However, cutting an edge removes its information flow, so we rebuild that path in two function-preserving steps. *Stage 1* attaches every new vertex in $N$ to its parent in $P$ (via an identity weight) and to all other parents with near-zero weights; this widens the layer while leaving the original activation almost unaffected. *Stage 2* reconnects each $v^n \in N$ to its former child in $C$, reinstating the cached weight $W[e_i]$, and links it to the remaining children with near-zero weights. The composition of these two stages equips the network with an extra non-linear transform and additional degrees of freedom that can be exploited during subsequent training.

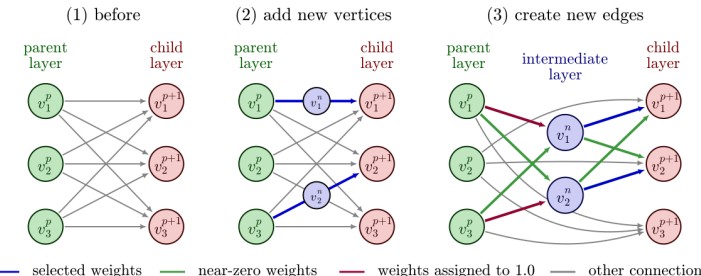

Figure 1: Expansion algorithm visualization. The blue vertices represent the intermediate vertices added during the expansion iteration.

**Stage 1: width expansion** $(P \rightarrow N)$. We create a fully–connected sub-layer whose weight matrix $W^{(1)} \in \mathbb{R}^{|N| \times |P|}$ is initialized as follows:

$$W_{ij}^{(1)} = \begin{cases} 1 & \text{if } v_j^p = \text{src}[v_i^n], \\ \varepsilon & \text{otherwise}, \end{cases} \quad \varepsilon \sim \mathcal{U}(0, 10^{-8}). \tag{4}$$

---

**Algorithm 1** Edge–wise expansion

---

1: **Input:** $G = (V_p, V_{p+1}, E, W)$, threshold $\tau_s$, empty sets $P, N, C$ and maps src, dst
2: Compute $g_{\mathrm{abs}}(e)$ for all $e \in E$
3: $E_{\mathrm{sel}} \leftarrow \{ e = (v^p, v^{p+1}) \in E \mid g_{\mathrm{abs}}(e) > \tau_s \}$
4: **for** $e = (v^p, v^{p+1}) \in E_{\mathrm{sel}}$ **do**
5:     Remove $e$ from $E$;  add new vertex $v^n$
6:     $\mathrm{src}[v^n] \leftarrow v^p, \mathrm{dst}[v^n] \leftarrow v^{p+1}$
7:     $P \leftarrow P \cup \{v^p\}, \ C \leftarrow C \cup \{v^{p+1}\}, \ N \leftarrow N \cup \{v^n\}$
8: **end for**
9: **for** $v^n \in N$ **do**
10:     **for** $v^p \in P$ **do**
11:       $E \leftarrow E \cup \{(v^p, v^n)\}$
12:       $W^{(1)}[(v^p, v^n)] \leftarrow \begin{cases} 1 & v^p = \mathrm{src}[v^n] \\ \varepsilon & \text{otherwise} \end{cases}$
13:     **end for**
14: **end for**
15: **for** $v^n \in N$ **do**
16:     **for** $v^{p+1} \in C$ **do**
17:       $E \leftarrow E \cup \{(v^n, v^{p+1})\}$
18:       $W^{(2)}[(v^n, v^{p+1})] \leftarrow \begin{cases} W[(\mathrm{src}[v^n], \mathrm{dst}[v^n])] & v^{p+1} = \mathrm{dst}[v^n] \\ \varepsilon & \text{otherwise} \end{cases}$
19:     **end for**
20: **end for**
21: **return** updated $G$

---

Thus, every original parent $v_j^p$ transmits its activation unchanged to its newly inserted child, while all other connections are initialized as numerically close to $0$. The near-zero uniform perturbation breaks the symmetry between the newly introduced parameters, allowing them to diverge during optimization without significantly altering the initial function of the layer.

**Stage 2: depth expansion** $(N \to C)$. We remove the selected edges, $E_{\mathrm{sel}}$, from $E$ and augment the parent set, $P \leftarrow P \cup N$, so that the network can expand recursively. The second fully–connected sublayer $W^{(2)} \in \mathbb{R}^{|C| \times |N|}$ is then added:

$$W_{ij}^{(2)} \ = \ \begin{cases} W[e_i] & \text{if } v_j^{p+1} = \mathrm{dst}[v_i^n], \\ \varepsilon & \text{otherwise}, \end{cases} \qquad \varepsilon \sim \mathcal{U}\big(0, 10^{-8}\big). \tag{5}$$

For the vertices not incident to edges from $E_{\mathrm{sel}}$, the connections do not change. During training, the near-zero parameters are free to expand, enabling the model to capture more expressive functions with minimal perturbation of the original solution.

**Forward pass.** Let $x \in \mathbb{R}^{|V_p|}$ be the input to the $p$-th layer, and write

$$x_P = \{ x_j \mid v_j^p \in P \}, \quad x_{\bar{P}} = \{ x_j \mid v_j^p \notin P \}.$$

Also let $W_{\mathrm{rem}} = W$ with all entries corresponding to edges $\bar{E} = E \setminus E_{\mathrm{sel}}$. Then the expanded layer produces

$$y \ = \ f \Big( \underbrace{W_{\mathrm{rem}} \, x_{\bar{P}}}_{\text{original path}} \oplus \underbrace{W^{(2)} \, f\big(W^{(1)} \, x_P\big)}_{\substack{\text{new two-step} \\ \text{expansion path}}} \Big), \tag{6}$$

where $W^{(1)} \in \mathbb{R}^{|N| \times |P|}, \quad W^{(2)} \in \mathbb{R}^{|C| \times |N|}, \ y$ is the output of the layer, $\oplus$ denotes concatenation. The inputs in $P$ flow through the new sublayer via $W^{(1)} \to f \to W^{(2)}$, while all other inputs follow the original transform $W_{\mathrm{rem}} \, x_{\bar{P}}$. If no edges are selected ($P = \varnothing$), then $W^{(1)}, W^{(2)}$ are empty and equation 6 reduces to the usual $y = f(Wx)$.

A schematic pseudocode implementation is presented in Alg. 1.

## 3.3 Training Strategy

**Training Procedure**. Our self-expanding network is trained with two main repeating events: layer expansion and edge pruning. The complete schedule is visualized in subsection A.1. We minimize the task loss

$$\arg\min_{f \in S} \mathbb{E}_{(x,y) \sim D}[L\big(f(x), y\big)], \tag{7}$$

where $S$ is a set of neural networks, $L$ is a loss function and $D$ is a dataset consisting of inputs $x$ and outputs $y$. At the end of every epoch we measure the validation loss $L_{\text{val}}$ and track its history $\{L_{\text{val}}^{(t-w)}, \ldots, L_{\text{val}}^{(t)}\}$ over a sliding window of $w$ epochs. If the average absolute change $\frac{1}{w}\sum_{k=0}^{w-1}|L_{\text{val}}^{(t-k)} - L_{\text{val}}^{(t-k-1)}|$ falls below a user-defined plateau threshold, $\delta$, and at least $T$ epochs have elapsed since the previous expansion, we perform the next expansion iteration. Once expansion occurs after $\Delta$ epochs, we prune edges with gradient metric values lower than the pruning threshold $\tau_p$ to stabilize the model size. We prune only the edges that were added during the expansion iterations, leaving other edges unchanged. Hyperparameter choice is detailed in subsection A.2.

**Network Pruning**. To enhance computational efficiency without compromising performance, we integrate structured pruning into SSONN training. We use the same gradient-based importance metrics from edge selection (Section 3.1) to identify pruning candidates. Pruning occurs gradually over $T_p$ epochs following each expansion, allowing new parameters sufficient training time to stabilize. The pruning threshold $\tau_p$ controls the removal rate, which varies by task but preserves model performance (in our experiments, up to $40\%$ of new edges could be removed without significant degradation). This selective pruning reduces computational costs in both training and inference.

## 4 Experiments

Our experimental framework utilizes Adam optimizer (Kingma & Ba, 2017) with learning rates ranging from $10^{-5}$ to $10^{-2}$ and weight decay from $10^{-4}$ to $10^{-2}$, selected through preliminary experiments for stable training. We evaluate batch sizes from $64$ to $256$ on NVIDIA GTX 1080 Ti GPU. Complete details of hyperparameter selection and optimization strategies are provided in subsection A.2.

The experiments were designed to evaluate the efficacy of the SSONN method in diverse scenarios, highlighting its ability to dynamically scale from minimal initial architecture while maintaining competitive accuracy and memory efficiency. All experiments began with a single fully-connected layer where the input dimension matches the number of features in each dataset and the output dimension corresponds to the number of target classes.

The algorithm proposed is implemented within the PyTorch framework. Due to the high sparsity of the final neural network architecture, the weights of the expanding neural network are stored as sparse matrices for memory efficiency. In Section 4.1, we compare our method on a wide variety of datasets with SOTA architectures in terms of accuracy and memory. In Section 4.2, we focus on the comparison with pruning.

### 4.1 Comparison with SOTA

MNIST and Fashion-MNIST Datasets are chosen as classical benchmarks for lightweight model evaluation. In addition, since our approach is centered on finding completely new architectures, the experiments also include datasets from Unseen NAS Datasets (Geada et al., 2024) including AddNIST, Language, MultNIST, CIFARTile, Gutenberg, Isabella, GeoClassing and Chesseract. Previous research devoted to neural architectural search (Ericsson et al., 2024) uses a common set of datasets and benchmarks. We follow the same experimental design. All these datasets evaluate the SSONN ability to dynamically adapt to unprecedented, heterogeneous tasks without architecture pre-tuning. The results of experiments on all datasets are presented below in Table 1 and Table 2.

As seen from the MNIST experiments 1, SSONN appears to be the most lightweight, holding the accuracy comparable to other architectures. On Fashion-MNIST 1 SSONN can demonstrate the same memory results only with MLP architectures. Regarding convolutional networks, the SSONN results are only comparable in terms of accuracy and number of parameters. Despite the lightweight

Table 1: Model comparison on MNIST (left) (Lecun et al., 1998) and Fashion-MNIST (right) datasets (Xiao et al., 2017). Our SSONN model achieves competitive performance with significantly fewer parameters compared to SOTA approaches. Training times for the proposed model were **598 seconds** on MNIST and **2177 seconds** on FashionMNIST. **Bold** indicates results that outperform others in accuracy or parameter efficiency.

| Model | Acc (%) | Parameters | Type |
|---|---|---|---|
| BMCNNwHFCs | **99.87** | 1 514 187 | CNN |
| VGG-5 | 99.69 | 3 646 000 | CNN |
| MLP | 99.25 | 2 900 000 | MLP |
| DNN-5 | 97.20 | 575 051 | MLP |
| SSONN (Ours) | 97.78 | **82 436** | MLP |

| Model | Acc (%) | Parameters | Type |
|---|---|---|---|
| Inception v3 | **94.44** | 27 161 264 | CNN |
| ResNet-18 | 92.28 | 11 689 512 | CNN |
| CTM-8000 | 91.50 | **527 250** | TM |
| SSONN (Ours) | 89.41 | 555 359 | MLP |

Table 2: Comparative analysis of model accuracy (%) and parameter counts on Unseen NAS Datasets (Geada et al., 2024). SSONN results (mean±3std) were obtained from 10 independent runs with different random seeds. All experiments were conducted with fixed hyperparameters. The benchmark compares standard architectures, random baseline and proposed SSONN (with separate parameter counts and training time). **Bold** indicates results that outperform others in accuracy or parameter efficiency. For NAS approaches, the number of parameters is not specified, because they select different architectures to suit the task. Training time (s) for SSONN includes both architecture expansion and model training.

| Model | Par. (M) | Gut. | Mult-N. | Add-N. | C-Tile | Chess. | Geo. | Lang. |
|---|---|---|---|---|---|---|---|---|
| ResNet-18 (He et al., 2015) | 11.7 | **49.98** | 91.55 | 92.08 | 45.56 | 57.83 | 80.33 | **97.00** |
| AlexNet (Alom et al., 2018) | 61.1 | 45.53 | 94.01 | 94.87 | 48.88 | 57.45 | 92.49 | 85.71 |
| VGG16 (Simonyan & Zisserman., 2014) | 138 | 44.00 | 90.43 | 92.06 | 24.43 | 55.69 | 93.67 | 84.54 |
| ConvNext (Liu et al., 2022) | 88.6 | 31.93 | 64.20 | 38.06 | 31.06 | 52.74 | 72.76 | 83.40 |
| MNASNet (Tan et al., 2018) | 4.4 | 38.00 | 87.70 | 90.51 | 48.49 | 56.26 | 86.00 | 84.63 |
| DenseNet (Huang et al., 2016) | 28.7 | 43.28 | 92.81 | 93.52 | 51.28 | 59.60 | 94.21 | 84.57 |
| ResNeXt (Xie et al., 2016) | 25.0 | 40.30 | 90.57 | 91.42 | 46.23 | 55.15 | 89.99 | 93.97 |
| PC-DARTS (Xu et al., 2019) | – | 49.12 | 96.68 | 96.60 | **92.28** | 57.20 | 94.61 | 90.12 |
| DrNAS (Chen et al., 2020) | – | 46.62 | **98.10** | 97.06 | 81.08 | 58.24 | **96.03** | 88.55 |
| Bonsai-Net (Geada et al., 2020) | – | 48.57 | 97.17 | **97.91** | 91.47 | 60.76 | 95.66 | 87.65 |
| DARTS (Liu et al., 2018) | – | 47.72 | 96.55 | 97.07 | 90.74 | 59.16 | 95.54 | 90.12 |
| Bonsai (Geada et al., 2020) | – | 29.00 | 39.76 | 34.17 | 24.76 | **68.83** | 63.56 | 76.83 |
| Random | – | 16.60 | 10.00 | 5.00 | 25.00 | 10.00 | 10.00 | 10.00 |
| SSONN (Acc) | – | 42.15 ±0.82 | 80.02 ±0.77 | 47.02 ±1.89 | 25.98 ±0.63 | 58.49 ±0.50 | 26.24 ±1.84 | 83.37 ±0.24 |
| SSONN params (M) | – | **0.72** ±0.12 | 5.72 ±0.71 | 9.35 ±0.82 | 0.62 ±0.09 | **1.34** ±0.39 | **1.1** ±0.24 | **0.69** ±0.11 |
| SSONN train time (s) | – | 473 | 5427 | 7894 | 343 | 529 | 1475 | 185 |

nature of SSONN, on most of Unseen NAS Datasets 2 the accuracy results are in the 2nd or 3rd place relative to the baselines. However, on some datasets, such as CIFARTile and GeoClassing, the proposed approach overfits very quickly and cannot show high accuracy due to the complexity of the problem. It is also worth noting that due to its implementation SSONN consumes relatively small GPU resources and less time for training.

## 4.2 COMPARISION WITH PRUNING

This section focuses on comparing the proposed approach and pruned neural networks. In each experiment shown in Table 3, we first trained and then pruned the SOTA architectures to the same accuracy as we did in SSONN by removing the weights with the smallest average gradient.

Through experimentation, on the MNIST and Fashion-MNIST datasets, SSONN proves to be a more lightweight solution than attempting to prune already trained architectures.

Table 3: Comparison of pruned architectures on MNIST and Fashion-MNIST datasets. We compare our SSONN model with pruned versions of standard architectures, where these models were pruned to match our parameter count while maintaining competitive accuracy. **Bold** indicates results that outperform others in accuracy or parameter efficiency

| | Original Models | | | Pruned Models | | | |
| | | | | MNIST | | F-MNIST | |
| Model | Par (M) | MNIST Acc (%) | F-MNIST Acc (%) | Par (M) | Acc (%) | Par (M) | Acc (%) |
|---|---|---|---|---|---|---|---|
| ResNet-18 | 11.18 | 99.01 | 91.95 | 2.80 | 97.31 | 1.13 | 89.38 |
| ResNeXt-50 | 23.00 | 98.70 | 91.71 | 5.75 | 97.72 | 0.87 | 89.10 |
| MNASNet-1 | 3.12 | 98.48 | 90.83 | 0.55 | 97.40 | 0.59 | 89.40 |
| EfficientNet-V2-S | 20.19 | 99.17 | 92.50 | 1.82 | 97.65 | 1.17 | 84.37 |
| MobileNet-V3-S | 1.53 | 98.54 | 90.11 | 0.10 | 97.54 | 0.11 | 89.29 |
| **SSONN (Ours)** | | | | **0.08** | **97.78** | **0.56** | **89.41** |

## 5 DISCUSSION AND LIMITATIONS

The comparison of SSONN with contemporary self-expanding frameworks is challenging. Existing self-expanding frameworks operate under a fundamentally different paradigm: they dynamically expand or modify architectures that are already built on existing state-of-the-art (SOTA) designs. For example, Firefly (Wu et al., 2021) and GradMax (Evci et al., 2022), reviewed in Section 2, prune or add layers within pretrained networks, inheriting their initial parameter counts and computational resources.

In contrast, SSONN starts from a trivial initial state (one linear layer) and expands only where necessary, eliminating reliance on SOTA templates. This distinction makes direct comparisons inequitable as competing methods benefit from architectural priors (inductive biases, pretrained components), while the proposed method constructs its topology entirely during training. Thus, direct memory and accuracy metrics fail to capture the unique advantage of SSONN to build efficient models ab initio without over-parameterization. Despite this fact, we performed the experiment presented in the GradMax paper to compare convergence rates on synthetic data using an elementary architecture for different approaches. The graphic of the experiment with its description can be found in subsection A.1.

The current implementation of SSONN is restricted to linear layers with ReLU activation function. This limitation stems from the framework design, requiring ReLU activation to preserve the network output upon initialization. As a result, the output remains unchanged in the forward pass after expansion. Combinations with convolutional layers or attention mechanisms have not yet been explored, limiting the method applicability to more complex architectures. However, in subsection A.4 experiments with large language models were conducted using SSONN. Additionally, unconstrained growth scenarios may lead to overfitting, which is mitigated through regularization techniques. More details on experimental results addressing these challenges are provided in Section 4.

## 6 CONCLUSION

This work introduces SSONN – Self-Scaled Optimized Neural Network, that constructs efficient architectures from a minimal initial state through reverse pruning. Using our method, training starts with a single fully-connected layer and gradually builds up the architecture for a specific task. Our experiment results validate the viability of the proposed paradigm for neural network design.

Limitations in architectural diversity and overfit problems highlight directions for future work focused on integrating domain-specific layer growth and theoretical analysis of convergence. We intend to add functionality for working with other types of layers such as convolution and attention layers. To solve the problem of overfit, we plan to implement automatic adjustment of hyperparameters during training. Further experiments on extending existing SOTA architectures with SSONN are required.

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

# A APPENDIX

## A.1 ABLATION STUDY

### A.1.1 EDGE METRIC COMPARISON

We evaluated SSONN's targeted expansion against two classes of baselines: (i) reversed-gradient criteria that expand at low-sensitivity edges, and (ii) magnitude-based criteria that select edges by $\ell_1$ or $\ell_2$ weight values. All models are trained on MNIST for $64$ steps, with a single expansion triggered at step $10$. Figure 2 presents the validation accuracy trajectories.

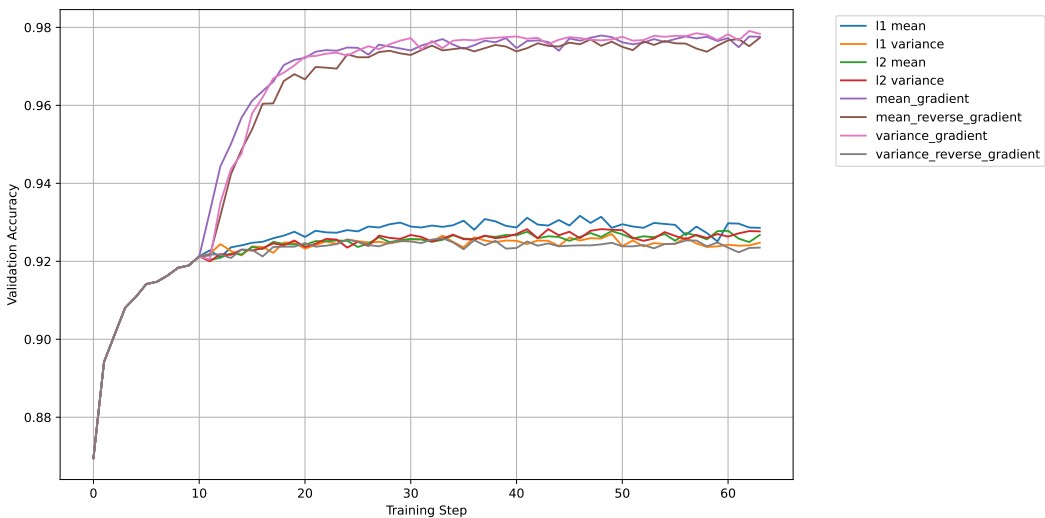

Figure 2: Validation accuracy for different metrics on MNIST

Immediately after the expansion event, both of the high-sensitivity criteria $g_{\text{abs}}$ and $g_{\text{var}}$ show a dramatic increase in precision from about $0.92$ to $0.98$. That indicates that the new capacity injected at the edges where the loss is most responsive produces substantial representational gains. In contrast, the reversed gradient baselines incur negligible improvement. Expanding at connections to which the loss is insensitive does not produce meaningful new learning directions. This failure is not just empirical but rooted in a theoretical inconsistency. Low-gradient edges correspond to near-flat regions of the loss surface, so appending sublayers there cannot break existing linearity; we formalize this argument in Subsection A.3.

Magnitude-based expansions ($\ell_1$ and $\ell_2$ weight norms) similarly underperform, delivering at best a $0.93$ plateau. Furthermore, gradient-targeted models converge more rapidly: by step $20$, the mean gradient variant surpasses $0.97$, while all other methods lag by at least $3\%$. These results confirm that SSONN's principle of injecting capacity at high-sensitivity edges is both theoretically justified and practically effective.

### A.1.2 COMPARISON WITH GRADMAX

Figure A.1.1 shows the training loss trajectories for SSONN and several competing methods, including GradMax (Evci et al., 2022). Unfortunately, we were unable to exactly reproduce their results from the available code. Instead, we ran an experiment in the same training regime for our model, using an MLP of a smaller size to demonstrate that any implementation advantage lies in the selection strategy, rather than the raw capacity.

GradMax (Evci et al., 2022) developing ideas from FireFly (Wu et al., 2021) starts with a 100:25:10 student architecture, aiming to achieve the results of a bigger pre-known 100:50:10 teacher architecture. Despite starting from a smaller network (100:10 MLP), SSONN achieves sub-unit loss in under $500$ steps, more than seven times faster than both GradMax and all other baselines, which only approach a loss of $1.0$ after $3{,}500$ steps. Thus, SSONN's targeted sublayer insertion at high-sensitivity

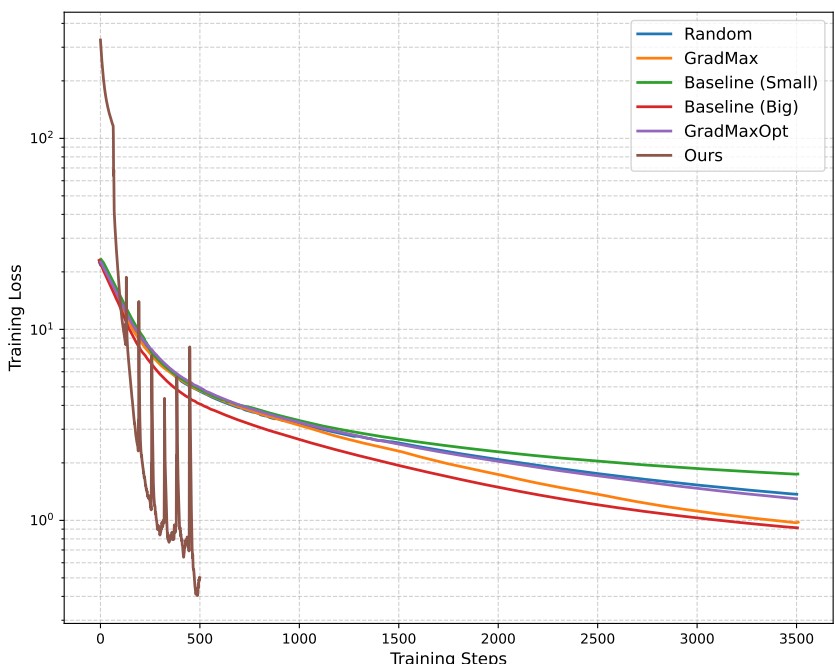

Figure 3: Comparison of SSONN and other expansion methods

edges unlocks representational power far more efficiently than existing criteria, even when those criteria are applied to larger models.

## A.2 HYPERPARAMETERS

The dynamic expansion in SSONN relies on four key hyperparameters that control the edge selection, expansion timing, and pruning processes. In the following, we describe the key components and their dataset-specific configurations (Table A.2).

**expansion_threshold** determines the threshold for selecting edges for expansion, based on the gradient-based importance metric $g_{abs}$ (Section 3.1). For complex tasks requiring larger neural networks, such as those in the Unseen NAS Datasets (Geada et al., 2024), we used lower values of `expansion_threshold` (approximately $0.1$) to allow more edges to be selected. For simpler tasks, such as MNIST, higher values (around $0.7$) were employed to limit expansion and maintain model compactness.

**pruning_threshold** defines the threshold for edge deletion during pruning, also based on the $g_{abs}$ metric. A value of $0.1$ was used in all experiments, as it effectively removed edges with minimal impact on model performance. This value was found to balance the reduction in model size with the preservation of task-specific accuracy, allowing up to $40\%$ of newly added edges to be pruned without significant performance degradation (Section 3.3).

**plateau_window_size** defines the number of epochs during which the validation loss history is analyzed to determine when expansion events occur. A fixed value of $5$ epochs was used, as it provided a robust window to assess whether the validation loss had stabilized. This choice ensured timely architecture expansion while avoiding unreasonable modifications.

**plateau_threshold** controls the sensitivity of the expansion trigger by specifying the maximum average absolute change in validation loss. The network is considered to have plateau if the performance, measured over a sliding window of length `plateau_window_size`, shows no meaningful improvement. When this condition is met, SSONN initiates a structural expansion of the architecture. Intuitively, lower values of `plateau_threshold` delay expansion, requiring more stable

convergence before modifying the architecture. Conversely, higher values permit earlier expansion, useful in tasks with noisy validation dynamics or high variance in early training.

Table 4: Dataset-specific hyperparameters for SSONN's dynamic expansion. The `expansion_threshold` controls edge selection via gradient-based importance, `plateau_threshold` determines expansion timing based on validation loss stability.

| Dataset | expansion_threshold | plateau_threshold |
|---|---|---|
| AddNIST | 0.30 | 0.05 |
| Chesseract | 0.05 | 0.50 |
| CIFArTile | 0.10 | 0.05 |
| GeoClassing | 0.40 | 0.50 |
| Gutenberg | 0.05 | 0.05 |
| Language | 0.20 | 0.05 |
| MultNIST | 0.30 | 0.005 |
| MNIST | 0.70 | 0.005 |
| FashionMNIST | 0.50 | 0.05 |

## A.3 ANALYTICAL JUSTIFICATION

**Edge sensitivity**. Let $L : \mathbb{R}^d \to \mathbb{R}$ be a differentiable loss function, $E$ is a set of edges, $e \in E$, $w_e \in \mathbb{R}$ is the weight of $e$, and $x \in X$ is a sample $x$ in training set $X$. If we assume

- that in global minima of $L$ $\forall e, x$ we have $g_{\text{abs}}(e, x) = 0$, where $g_{\text{abs}}$ is defined following formula 1,
- for $L$, if $\left| \frac{\partial L(x_2)}{\partial x} \right| > \left| \frac{\partial L(x_1)}{\partial x} \right|$, $x_1 \neq x_2$, then $L(x_2) > L(x_1)$.

then the replacement of the edge with the largest $\sum_x g_{\text{abs}}(e, x)$ in local minima of $L$ is the optimal way to decrease loss by network expansion.

Let's define $e_1 \in E$ with associated weights $w_1(e)$ and $e_2 \in E$ with associated weights $w_2(e)$, and $w_1 \neq w_2$. If $\sum_{x,e_1} g_{\text{abs}}(e_1, x) > \sum_{x,e_2} g_{\text{abs}}(e_2, x)$ then $\sum_x L(e_1, x) > \sum_x L(e_2, x)$.

Let's split all the edges to $k$ subgroups, each subgroup is related to sublayer. Loss derivative is computed via chain rule and is a product: $\frac{\partial L}{\partial x} = \frac{\partial L}{\partial w(e^k)} \cdot \frac{\partial w(e^k)}{\partial w(e^{k-1})} \cdot \dots \cdot \frac{\partial w(e^2)}{\partial w(e^1)} \cdot \frac{\partial w(e^1)}{\partial x}$

Using the property if $|\prod_k a_k| > |\prod_k b_k|$ then $\sum_k |a_k| > \sum_k |b_k|$ and our assumption about properties of the $L$ it's easy to show correctness of the lemma.

We select the edge (e) with the maximum sum $\sum_x g_{\text{abs}}(e, x)$, as it has the greatest contribution to the loss gradient.

Adding parameters to such an edge maximizes the reduction of (L), since:

$$\Delta L \propto \sum_x g_{\text{abs}}(e, x).$$

From Lemma, it follows that replacing an edge with a greater sum of gradients leads to a larger reduction in (L).

**Computational and memory cost of the gradient-based metrics.** Let $B = |dataloader|$ be the number of mini-batches processed, $E$ — the number of learnable weights (edges) in the target layer, $C_{\text{fb}}$ — the computational cost of one forward plus backward pass through the network on a single mini-batch.

Following both Equation 1 and Equation 2, algorithm steps are:

1. one forward+backward pass per batch,
2. one element-wise addition of two $E$-length tensors per batch,

3. a single post-loop reduction and normalization over $E$ elements.

Hence the *time* complexity is

$$T_{g_{\mathrm{abs}}} = \mathcal{O}\big(B\,(\,C_{\mathrm{fb}} + E\,)\big) = \mathcal{O}\big(B\,C_{\mathrm{fb}}\big), \tag{8}$$

because $C_{\mathrm{fb}} \gg E$ in most settings.

The *additional memory* is dominated by the accumulated values in tensors: $M_{g_{\mathrm{abs}}} = \mathcal{O}(E)$ for both mean and variance metrics. No other step stores intermediate tensors larger than the edge-metric itself, so the overall RAM overhead is linear in the layer size and independent of the number of batches.

## A.4   COMPARISON WITH DISTILLATION

This section evaluates the adaptability of SSONN by integrating it into established architectures through knowledge distillation of specific blocks within large language models. These experiments are conducted based on the hypothesis that SSONN can improve model performance by effectively leveraging existing feature representations.

Due to the resulting architectures, SSONN can be used for distillation and calibration of individual MLP model blocks. One common task suitable for this scenario is the distillation of large MLP blocks in LLM. Table 5 presents distillation results for critical MLP blocks (layers 7, 14, and 21) of the Qwen 2.5 model (Qwen et al., 2025) on the MMLU dataset (Hendrycks et al., 2020). These layers were identified as most influential based on an ablation study. The table reports the performance of the original model, the model with these layers removed, and the model with the layers distilled using both a linear layer and SSONN. As a result of distillation, ssonn demonstrated better accuracy, reducing the number of parameters in each selected layer by approximately 2 times.

Table 5:  MMLU benchmark results for Qwen 2.5 (Qwen et al., 2025) after distilling its MLP blocks. Models were distilled with layers 7, 14, and 21. We compare three strategies: full layer distillation to a linear layer, our proposed SSONN distillation, and simple layer removal. Results are shown for the full MMLU dataset (Hendrycks et al., 2020) and its mathematics subsets.

| Setup | MMLU Acc (%) | MMLU abstract algebra Acc (%) | MMLU college mathematics Acc (%) |
|---|---|---|---|
| Original Model | 42.17 | 35.02 | 38.43 |
| MLP distillation | 40.87 | 33.50 | 36.87 |
| No layer | 39.42 | 29.81 | 32.49 |
| SSONN distillation | **43.98** | **43.76** | **51.02** |

