# OpenReview forum: "SSONN: Self-Scaled Optimized Neural Network"
_ICLR.cc/2026/Conference — Submitted to ICLR 2026_

### Official Review · Reviewer_25qV · 2025-10-28

**Soundness:** 2
**Presentation:** 2
**Contribution:** 1
**Rating:** 0
**Confidence:** 5

**Summary:**

The paper proposes a process for progressively building the architecture of feedforward (dense) neural networks. This process starts with a simple network (single fully-connected layer) and gradually expands the architecture by adding sublayers. The method is evaluated on public datasets and shows some advantages against baselines. However, the authors have not considered methodologies which are the direct competition to the proposed approach. Claims such as "We propose a novel expansion-centric methodology that challenges the traditional train-then-compress paradigm" are not correct, as similar approaches to the one in this paper have been proposed several years ago.

**Strengths:**

- The proposed method shows some advantages against baselines on small-dimensional datasets.

**Weaknesses:**

- The text includes unfounded claims
 - A part of the literature solving the same problem is not considered. These include Progressive Operational Networks (POPs), Heterogeneous Multilayer Generalized Operational Perceptrons (HeMLGOPs), Operational Neural Networks (ONNs), Self-organized ONNs (SelfONNs)
 - No comparisons are provided with the directly relevant methods.
 - The proposed method is suitable only for dense networks, thus is limited only for low-dimensional data.

**Questions:**

- How does the method compare with the actual competition?
 - How does the proposed method compare with baselines and existing related methods in terms of training speed?
 - How to choose the hyper-parameters of the method for new datasets without requiring tedious try-and-error through grid search?

---

### Official Review · Reviewer_4qXU · 2025-10-29

**Soundness:** 2
**Presentation:** 2
**Contribution:** 1
**Rating:** 2
**Confidence:** 4

**Summary:**

The paper introduces SSONN, a neural-network training approach that starts from a minimal network and dynamically adds or removes edges during learning. The model expands only when gradients indicate insufficient capacity. High-sensitivity connections are expanded into small sublayers (width and depth), while redundant ones are pruned. Each expansion is initialized to preserve the current function under ReLU, so the model grows without disrupting learned behavior.

**Strengths:**

1. The idea of dynamically growing a network based on gradient-driven edge importance is simple and straightforward. The function-preserving initialization under ReLU is a neat trick to avoid loss spikes during expansion.

2. The paper is clearly written.

**Weaknesses:**

1. The evaluation is confined to small datasets such as MNIST, Fashion-MNIST, and Unseen NAS benchmarks. These are toy-scale tasks that do not convincingly demonstrate the method’s effectiveness or scalability to modern large-scale problems (ImageNet or transformer-based models)

2. In Table 1, the paper compares SSONN mainly with larger models that tend to overfit on simple datasets like MNIST and Fashion-MNIST. Fairer baselines such as ResNet-20 or ResNet-56 (similar parameter scale) are missing, making it unclear whether the gains come from the method itself or differences in model size.

3. The paper lacks evaluation against recent structured pruning, knowledge distillation, and efficient NAS methods such as [1,2,3], making it difficult to position SSONN within the current landscape.

4. While the paper claims parameter and compute efficiency, it does not consistently report training time, inference time, memory footprint, or FLOPs across models. This prevents a fair assessment of the claimed efficiency gains.

5. The proposed method is only demonstrated on fully connected MLPs and relies on assumptions that may not generalize to CNNs or Transformers.

[1] Fang, Gongfan, et al. "Depgraph: Towards any structural pruning." Proceedings of the IEEE/CVF conference on computer vision and pattern recognition. 2023.

[2] Dong, Peijie, Lujun Li, and Zimian Wei. "Diswot: Student architecture search for distillation without training." Proceedings of the IEEE/CVF Conference on Computer Vision and Pattern Recognition. 2023.

[3] Li, Guihong, et al. "Zero-shot neural architecture search: Challenges, solutions, and opportunities." IEEE Transactions on Pattern Analysis and Machine Intelligence 46.12 (2024): 7618-7635.

**Questions:**

Please refer to weeknesses

---

### Official Review · Reviewer_MRFm · 2025-10-31

**Soundness:** 2
**Presentation:** 3
**Contribution:** 1
**Rating:** 4
**Confidence:** 4

**Summary:**

This paper introduces the Self-Scaled Optimized Neural Network (SSONN), a dynamic network growth method that challenges the conventional "train-then-compress" paradigm. Instead of pruning a large, over-parameterized model, SSONN starts with a minimal architecture (a single linear layer) and iteratively expands its complexity during training. The expansion is guided by a gradient-based importance metric, $g_{abs}$ (mean absolute gradient), which identifies "critical" edges. These selected edges are replaced by new intermediate neurons, effectively creating new sublayers. The method aims to build compact, task-specific architectures from scratch, reducing initial computational costs. Experiments on MNIST, Fashion-MNIST, and the Unseen NAS datasets demonstrate that SSONN can achieve comparable accuracy to baseline models while using significantly fewer parameters.

**Strengths:**

* The paper proposes an expansion-centric methodology ("reverse pruning") as an alternative to the dominant pruning and distillation paradigms. This "grow-from-scratch" concept is an interesting direction for efficient model design.
* On the tested benchmarks (MNIST, Fashion-MNIST), the method demonstrates high parameter efficiency, achieving competitive accuracy with substantially fewer parameters compared to both standard and pruned SOTA models.

**Weaknesses:**

1.  **Limited Applicability:** The method's most critical weakness is its restriction to linear layers and the ReLU activation function. The paper states this is necessary to preserve the network's output function immediately after expansion. This limits the method's applicability and prevents its use in most modern, high-performance architectures that rely on convolutional layers, attention mechanisms, and alternative activations (e.g., GeLU, SiLU, SwiGLU).
2.  **Limited Experimental Scope:** The empirical evaluation is constrained to simple, small-scale datasets. The paper notes that SSONN overfits on some Unseen NAS datasets (CIFARTile, GeoClassing), raising questions about its generalization capabilities on more difficult tasks.
3.  **Missing Practical Efficiency Metrics:** The paper's claims of efficiency are based almost entirely on parameter counts. This is an incomplete picture. The review lacks crucial practical metrics:
    * **Memory Footprint:** There is no reporting on peak training or inference VRAM usage.
    * **Latency:** Inference speed (e.g., in ms) is not reported. A model with fewer parameters but a highly sparse and irregular structure (as SSONN seems to produce) may not be faster in practice on modern GPUs optimized for dense matrix operations.
    * **Method Overhead:** The computational overhead of the $g_{abs}$ metric (requiring $O(E)$ storage for all edge gradients) is not empirically measured.
4.  **Preliminary LLM Analysis:** The experiment in Appendix A.4, while interesting, is preliminary. SSONN is used to distill LLM MLP blocks and is compared only against trivial baselines ("No layer" and "MLP distillation" to a single linear layer). To be convincing, this analysis should compare SSONN against established, specialized methods for LLM component pruning or compression.
5.  **Limited Theoretical Analysis:** The "Analytical Justification" in Appendix A.3 is more of a high-level justification than a formal analysis. It relies on simplifying assumptions and does not provide rigorous proof of convergence or optimality for the proposed expansion strategy.

**Questions:**

1.  The restriction to the ReLU activation function is a primary limitation. Could the output-preserving expansion mechanism be adapted for other non-linearities, such as GeLU or SwiGLU, which are standard in modern Transformer models? If not, does this fundamentally prevent SSONN from being a general-purpose method for efficient architecture design?
2.  Could the authors provide practical efficiency metrics beyond parameter counts? Specifically, what is the peak VRAM usage during training (including the $g_{abs}$ metric storage) and the average inference latency (in milliseconds) on a GPU, compared to the dense baselines in Tables 1 and 3?
3.  For the LLM experiment in Appendix A.4, how does SSONN compare against established structured or unstructured pruning methods specifically designed for FFN blocks in Transformers? The current baselines of "No layer" and "MLP distillation" are not representative of SOTA compression techniques.
4.  The paper notes overfitting on more complex Unseen NAS datasets. Does the gradient-driven expansion strategy inherently create architectures with high variance that struggle to generalize? How is this overfitting risk managed beyond the simple pruning of newly added edges?

---

### Official Review · Reviewer_wxEu · 2025-11-01

**Soundness:** 3
**Presentation:** 3
**Contribution:** 3
**Rating:** 6
**Confidence:** 4

**Summary:**

The paper introduces a self-expanding method designed to eliminate the traditional train–then–compress paradigm of lightweight model design. Instead of starting with an over-parameterized model, SSONN begins with a single linear layer and expands dynamically based on gradient-driven edge importance metrics. The key innovation lies in the reverse pruning strategy, where new sublayers are added only to high-sensitivity regions, using a combination of gradient magnitude and variance as triggers.

Experiments across MNIST, Fashion-MNIST, and the Unseen NAS Datasets show that SSONN can match or approach higher accuracy while using up to 10× fewer parameters. Ablation studies confirm that gradient-based edge expansion outperforms weight-based or random expansion. The paper also includes theoretical analysis, pruning comparisons, and a brief exploration of distillation use in large language models (e.g., Qwen 2.5).

**Strengths:**

SSONN proposes a shift from compression to growth: starting small and expanding dynamically as training progresses. This inversion is well-motivated for edge devices and resource-constrained deployment, where initial over-training is impractical. The authors make a convincing case that adaptive growth can lead to more sustainable model design.

The method is logically defined and easy to follow. The pipeline — consisting of edge selection, sublayer insertion, and pruning — is presented with complete algorithmic pseudocode .

By initializing new sublayers with near-zero perturbations and identity mappings, SSONN maintains the function continuity of the model after each expansion, avoiding catastrophic loss spikes. This resembles function-preserving transformations in Net2Net, but SSONN achieves it dynamically through local gradient triggers instead of predefined architecture templates.

**Weaknesses:**

The current implementation only supports fully connected layers with ReLU activation. This restriction significantly narrows practical applicability, as modern vision and language models depend on convolutional and attention mechanisms. Although the authors acknowledge this in Section 5 and 6, the lack of even a small CNN or transformer demonstration weakens the claim of architecture independence.

On Fashion-MNIST, SSONN  89.4% vs Inception v3 94.4% and ResNet-18 92.3%. On complex Unseen NAS datasets (e.g., CIFARTile, GeoClassing), accuracy is mid-range (about 25–58%) compared to NAS baselines ( larger than 90%). These gaps suggest that current gradient-based expansion alone may not provide sufficient expressivity for high-dimensional or spatial tasks.

Although claimed to be efficient, the paper does not quantify FLOPs or memory overhead during dynamic expansion. The cost of computing gradients for all edges (needed for $g_{abs}(e)$) could scale poorly with large networks. Without such an analysis, it’s uncertain whether SSONN remains computationally feasible on large-scale data.

Comparisons against pruning and NAS baselines are not fully balanced in compute or training budget:
- NAS methods (PC-DARTS, DrNAS) are inherently large-scale and may not be directly comparable.
- SSONN’s advantage in parameter count is offset by possible differences in training duration or optimizer setup.
A fairer comparison would control for training epochs, GPU hours, or validation accuracy levels.

A notable omission in the related work section is the absence of discussion on modern NAS frameworks specifically designed for memory efficiency, which are directly relevant to the SSONN objective of lightweight, resource-constrained model construction. For instance, MemNAS and TinyNAS or  GreedyNASv2 offer explicit formulations for low-memory neural architecture optimization.
In contrast, SSONN’s independence from NAS search spaces is a distinguishing feature, but without acknowledging these state-of-the-art NAS-based efficiency strategies, the positioning of SSONN’s contribution appears incomplete.

Conceptually, SSONN continues the line of gradient-driven or curvature-driven expansion GradMax (Evci et al., 2022) and SENN (Mitchell et al., 2024). While the method simplifies and unifies these ideas, it does not introduce a fundamentally new theoretical principle.

Minor presentation and clarity issues: Some redundancy exists in Introduction and Related Work; Grammar and phrasing occasionally non-idiomatic (“consumes relatively small GPU resources”).

**Questions:**

A simple experiment on CIFAR-10 with convolutional expansion would strengthen the paper considerably.

---

### Meta-Review · Area_Chair_8wfM · 2026-01-05

**Summary:**

This paper proposed a self-scaled optimized neural network for neural architecture design. Different from training then pruning method, it begins with a single linear layer and add nodes and connections only to critical places to improve the accuracy of the network. Experiments on MNIST, Fashion-MNIST and Unseen NAS datasets demonstrates the proposed method outperforms the training then pruning method.

Reviewers acknowledge that the paper is clearly written and have good results on the tested benchmarks. The major concerns from reviewers that informed the rejection of this paper are:
1. This paper only considers linear layer with ReLU activation, which limits its applications.
2. This paper only shows the number of parameters of the designed network, other important factors such as FLOPS, memory consumption and so on are not discussed.
3. Benchmark used (MNIST, Fashion-MNIST, Unseen NAS) are too simple, more difficulty tasks are needed to demonstrate the effectiveness of the method.

**Reviewer Concerns:**

As there is no rebuttal from the authors, the reviewers concerns mentioned above are not addressed.

**Reviewer Scores:**

This paper get scores of 6, 4, 2, 0. As no rebuttal is posted, I believe the reviewers will remain their scores.

---

### Decision · Program_Chairs · 2026-01-26

Reject